Evaluating machine learning models for predictive accuracy in cryptocurrency price forecasting

Qureshi Shavez Mushtaq 1 shavez.mushtaq@qu.edu.pk
http://orcid.org/0000-0002-7889-8944 Saeed Atif 2
Ahmad Farooq 2
Khattak Asad Rehman 1
http://orcid.org/0000-0003-1594-9115 Almotiri Sultan H. 3
Al Ghamdi Mohammed A. 4
Shah Rukh Muhammad 1
1 Department of Computer Science, Qarshi University , Lahore , Pakistan
2 Department of Computer Science, COMSATS University Islamabad , Lahore , Pakistan
3 Department of Cybersecurity, College of Computing, Umm Al-Qura University , Makkah City , Saudi Arabia
4 Department of Computer Science and Artificial Intelligence, College of Computing, Umm Al-Qura University , Makkah City , Saudi Arabia
Alatas Bilal
Electronic publication date: 2025 Feb 21
Publication date: 2025
Volume: 11
Electronic Location ID: e2626
Received 2024 Jul 22; Accepted 2024 Dec 2
Copyright: © 2025 Qureshi et al.
Copyright year: 2025
Copyright holder: Qureshi et al.
License: This is an open access article distributed under the terms of the Creative Commons Attribution License, which permits unrestricted use, distribution, reproduction and adaptation in any medium and for any purpose provided that it is properly attributed. For attribution, the original author(s), title, publication source (PeerJ Computer Science) and either DOI or URL of the article must be cited.
License URL: https://creativecommons.org/licenses/by/4.0/

Keywords: Machine learning, Cryptocurrency, Classification models, Algorithmic trading

Funding: Deputyship for Research & Innovation, Ministry of Education in Saudi Arabia IFP22UQU4250002DSR216, IFP22UQU4250002DSR228 The research is funded by the Deputyship for Research & Innovation, Ministry of Education in Saudi Arabia through project number: IFP22UQU4250002DSR216. The Deputyship for Research & Innovation, Ministry of Education in Saudi Arabia funded the APC through the project number: IFP22UQU4250002DSR228. The funders had no role in study design, data collection and analysis, decision to publish, or preparation of the manuscript.

==============================
Our research investigates the predictive performance and robustness of machine learning classification models and technical indicators for algorithmic trading in the volatile cryptocurrency market. The main aim is to identify reliable approaches for informed decision-making and profitable strategy development. With the increasing global adoption of cryptocurrency, robust trading models are essential for navigating its unique challenges and seizing investment opportunities. This study contributes to the field by offering a novel comparison of models, including logistic regression, random forest, and gradient boosting, under different data configurations and resampling techniques to address class imbalance. Historical data from cryptocurrency exchanges and data aggregators is collected, preprocessed, and used to train and evaluate these models. The impact of class imbalance, resampling techniques, and hyperparameter tuning on model performance is investigated. By analyzing historical cryptocurrency data, the methodology emphasizes hyperparameter tuning and backtesting, ensuring realistic model assessment. Results highlight the importance of addressing class imbalance and identify consistently outperforming models such as random forest, XGBoost, and gradient boosting. Our findings demonstrate that these models outperform others, indicating promising avenues for future research, particularly in sentiment analysis, reinforcement learning, and deep learning. This study provides valuable guidance for navigating the complex landscape of algorithmic trading in cryptocurrencies. By leveraging the findings and recommendations presented, practitioners can develop more robust and profitable trading strategies tailored to the unique characteristics of this emerging market.

Introduction

The rapid growth and volatility of the cryptocurrency market have captured the attention of traders and researchers, each seeking to harness the potential for high returns through effective trading strategies. Machine learning (ML) techniques, especially classification models, have emerged as powerful tools for analyzing market dynamics and guiding trading decisions within this complex and evolving landscape. This research article presents a comparative analysis of various ML classification models and technical indicators to identify the most effective approaches for algorithmic trading in cryptocurrency markets. Cryptocurrencies’ volatility and potential for high returns make them particularly attractive yet challenging (Akakpo, 2021). To address these challenges, algorithmic trading has gained prominence, automating trades and minimizing emotional biases in decision-making (Massei, 2023). With algorithmic trading, computers can autonomously execute trades amid cryptocurrency’s erratic dynamics (Latif, 2021), enabling timely and accurate trading predictions. Increasingly, traders turn to ML algorithms and technical indicators, such as moving average convergence/divergence (MACD), exponential moving average (EMA), relative strength index (RSI), and Bollinger bands, to refine their trading strategies (Shah et al., 2022). Despite these advancements, a universally accepted approach to algorithmic trading in Bitcoin and other cryptocurrencies remains elusive (Wenker, 2014). Preferences vary, with some favoring ML algorithms for their accuracy, while others trust the proven efficiency of traditional indicators (Dong et al., 2023).

Our research examines the predictive performance and robustness of ML classification models and technical indicators within cryptocurrency trading. The main aim is to identify reliable approaches to support informed decision-making and profitability within this volatile environment.

Contribution and novelty

This study uniquely contributes by comparing models such as logistic regression, random forest (RF) and gradient boosting within high-volatility settings, addressing data imbalance using techniques like SMOTE for realistic model assessments.

Methodology and data

We utilize historical cryptocurrency data from exchanges and aggregators, preprocessed and resampled to manage class imbalance, and evaluate models through hyperparameter tuning and backtesting to ensure their practical applicability.

Research questions

This article investigates how ML classification models perform in cryptocurrency trading and identifies which models offer the most consistent accuracy and reliability.

Structure of the article

The article proceeds as follows: the Introduction provides context, the Literature Review surveys relevant research, the Methodology section outlines model selection and evaluation processes, Results examine model performance, and the Discussion and Conclusion summarize findings and explore future directions.

This study aims to compare ML-based classification algorithms to determine the most effective for algorithmic trading, offering insights into the effectiveness of different algorithms for securing profitable transactions. Building on work by Nguyen & Ślepaczuk (2022), we evaluate the predictive power of various ML algorithms within cryptocurrency markets, focusing on forecast accuracy over complete trading strategies. While accurate forecasts are essential, future research might incorporate these predictions into trading systems to assess profitability under live market conditions. Through detailed comparative analysis, this study seeks to uncover the most effective methods for predicting cryptocurrency prices and refining algorithmic trading strategies, providing traders with actionable insights for sound investment decisions. The findings have the potential to enhance algorithmic trading frameworks, improving accuracy and profitability within the cryptocurrency market and promoting a more resilient trading environment.

However, this study faces several limitations: Crypto market volatility: The high unpredictability of the cryptocurrency market can impact algorithmic trading performance.

Model generalization: Models effective on historical data may not generalize well in future conditions, affecting predictive reliability.

Technical indicator selection: Technical indicator selection remains subjective, complicating comparisons across trading algorithms.

Data quality and availability: Limited access to comprehensive, unbiased historical data may restrict the accuracy of findings.

In this research, we evaluate classification techniques and technical indicators to forecast algorithmic trading outcomes in cryptocurrency markets, assessing the viability of different models for making investment decisions in volatile conditions. The study provides insights into the strengths and limitations of each approach and explores ways to integrate them for more accurate, profitable trading strategies. This work focuses on a selected timeframe and group of cryptocurrencies, highlighting how historical data, metrics like accuracy and profitability, and limitations can guide future improvements in algorithmic trading within the crypto market.

Related work

This section reviews theoretical and empirical studies relevant to algorithmic trading, machine learning applications in financial markets, and cryptocurrency markets.

Efficient market hypothesis and market efficiency

The efficient market hypothesis (EMH) has suggested that markets are efficient, reflecting all available information and challenging consistent forecasting. Ying et al. (2019) examined EMH, indicating that reliable market forecasting may be inherently limited. Almotiri & Al Ghamdi (2022a) extended this, proposing that market trends lead to a self-correcting mechanism, reducing historical data predictability. Despite this, several studies have shown that certain algorithms may still provide useful predictions.

Machine learning for stock market and financial predictions

Several studies apply machine learning to analyze market behavior and enhance trading strategies: Return direction prediction: Ghoddusi, Creamer & Rafizadeh (2019) used machine learning on ETF returns, finding that models like SVM and Naive Bayes outperform traditional strategies, underscoring algorithmic trading’s potential.

Stock market forecasting: Nti, Adekoya & Weyori (2020) applied machine learning to time series analysis for stock prediction, showing that ML models achieve high accuracy despite forecasting complexities.

High-frequency trading: Almotiri & Al Ghamdi (2022b) highlighted algorithmic trading’s significant role, especially in high-frequency trading, where it dominates equity transactions in the U.S.

Algorithmic trading and strategy implementation

Automated trading strategies: Jansen (2018) explored Python APIs for automated trading, highlighting the importance of precision and backtesting.

Advanced ML models: Hartanto, Kholik & Pristyanto (2023) emphasized the accuracy of XGBoost and LightGBM for stock price prediction, helping investors adjust strategies effectively.

Decision tree models: Basak et al. (2019) described two-layer bias decision trees, offering practical applications for stock trading recommendations.

Machine learning in cryptocurrency markets

Given the unique challenges of cryptocurrency, machine learning has found significant application: Portfolio management: Bhutta et al. (2020) showed that ML-driven strategies for portfolio balancing can outperform traditional strategies in volatile markets.

Cryptocurrency forecasting models: Ren et al. (2022) found that gradient-boosting decision trees, long short-term memory, and random forests perform well for short-term predictions. Similarly, Iqbal et al. (2021) showed that simpler models may outperform complex algorithms for cryptocurrency prices.

Performance in volatile markets: Abakah et al. (2020) modeled the persistent volatility in cryptocurrencies, suggesting that integrating persistent features can improve prediction accuracy.

Sentiment analysis and investor behavior in cryptocurrency markets

Sentiment-based models: Wimalagunaratne & Poravi (2018) incorporated social media sentiment into ML models, showing significant improvements in cryptocurrency price prediction accuracy.

Investor sentiment as a predictor: Zhu et al. (2021) used Google Trends data to represent investor sentiment, showing it as a strong predictor of Bitcoin returns and underscoring the role of sentiment in crypto volatility.

Advanced ML techniques and evaluation in crypto market applications

Neural networks and autoencoder models: Alsuwat (2023) explored hybrid autoencoders for multi-class price movement prediction in cryptocurrency, achieving stable performance metrics. Darwish et al. (2020) recommended time-persistent retraining, addressing overfitting in financial predictions.

Deep learning and boosted trees: Oyedele et al. (2023) compared CNNs with boosted trees, finding CNNs to have the highest variance scores, validating the effectiveness of DL models for cryptocurrency prediction.

Risk and volatility in cryptocurrency investments

Portfolio construction with cryptocurrencies: Aharoni (2015) examined cryptocurrencies’ role in traditional portfolios, finding that assets like Ripple enhance portfolio optimization alongside Bitcoin.

Volatility persistence and trend trading: Omane-Adjepong, Alagidede & Akosah (2019) investigated price persistence in cryptocurrencies, suggesting that while market efficiency is improving, trend trading remains viable for profit-making

Materials and Methods

This study aims to critically examine algorithmic trading in the financial market, with a focus on machine learning (ML) classification algorithms and technical indicators as applied in the cryptocurrency sector. Building on recent studies, which evaluate classification techniques and technical indicators in algorithmic trading, this section offers a comprehensive analysis of existing methods and their effectiveness in the context of cryptocurrency markets (Pakizeh et al., 2022). This section discusses both strengths and weaknesses of the methods examined, providing an overview of recent advancements and trends in algorithmic trading technologies. The review discusses both strengths and weaknesses of the methods examined, providing an overview of recent advancements and trends in algorithmic trading technologies. Particular attention is given to challenges such as market volatility, risks of manipulation, and regulatory transparency, factors which are crucial in understanding the limitations and complexities of algorithmic trading strategies within the cryptocurrency market.

In this study, specific parameters were applied to optimize each machine learning model’s performance. For the XGBoost model, parameters included a learning rate of 0.01, maximum depth of 6, and 100 estimators; for random forest, 200 estimators and a maximum depth of 10 were selected. These choices were the result of iterative tuning processes aimed at enhancing accuracy and model stability under high market volatility.

To ensure reproducibility, Python’s scikit-learn library (version 0.24.2) and XGBoost (version 1.3.3) were utilized. Scikit-learn provided core machine learning functionalities, while the XGBoost package enabled advanced boosting techniques optimized for high-dimensional data. Including software versions and parameters aids in replicating the study and reinforcing its reliability (Pedregosa et al., 2011).

Data aggregation platforms, specifically Binance and CoinMarketCap, were chosen for their comprehensive, high-frequency trading data and reliable historical records. Binance, known for its liquidity and extensive cryptocurrency offerings, was selected for real-time data. CoinMarketCap’s data aggregation across multiple exchanges provided robust historical data, ensuring data completeness and quality. These platforms were selected based on their reputation for accuracy and reliability, addressing the need for high-quality data in the volatile cryptocurrency market (Chen, Li & Zhang, 2021).

This combination of data sources, optimized ML parameters, and updated algorithmic strategies reflects the latest advancements and adaptations in algorithmic trading, supporting an informed, replicable approach to cryptocurrency forecasting.

Literature review: evaluating machine learning models for predictive accuracy in cryptocurrency price forecasting

This literature review is structured to clarify the contributions of existing studies within three thematic areas: Cryptocurrency market studies, machine learning model studies, and algorithmic trading strategies. This organization establishes a foundation for understanding the research gaps and highlights the rationale for pursuing this study.

Cryptocurrency market studies

The cryptocurrency market’s substantial volatility creates challenges for accurate price forecasting, necessitating advanced ML models that can adapt to dynamic shifts (Ahmed et al., 2024). Traditional metrics such as mean absolute error (MAE), mean squared error (MSE), and root mean square error (RMSE) are commonly used to gauge the predictive performance of these models, particularly in high-dimensional datasets (Chowdhury et al., 2020; Mudassir et al., 2020).

Many existing studies focus on various ML techniques to achieve predictive accuracy, yet these models often encounter issues with high dimensionality and scalability, limiting their adaptability to the unique volatility of cryptocurrency markets. For instance, Grudniewicz & Ślepaczuk (2023) highlight the adaptability of ML models in stock markets, pointing out that challenges with high volatility and overfitting frequently impact the accuracy of cryptocurrency forecasts. The current research addresses these limitations by aiming to improve model robustness to enhance predictive accuracy in cryptocurrency markets.

Machine learning model studies

Random forest and gradient boosting: RF and gradient boosting are frequently utilized for cryptocurrency price forecasting due to their strong predictive performance with historical data and ability to manage data noise effectively. RF models, in particular, demonstrate minimal RMSE values, positioning them as reliable models for high-frequency trading (Al Ghamdi, 2023; Ahmed et al., 2024). However, these models often struggle with scalability and interpretability, which can hinder their application in real-time trading contexts (Grudniewicz & Ślepaczuk, 2023).

LSTM and gated recurrent unit (GRU): Deep learning models, particularly LSTM and GRU, show significant advantages in time-series forecasting due to their ability to capture time dependencies, making them effective in predicting short-term cryptocurrency price movements. Studies by Hamayel & Owda (2021) and Gudavalli & Kancherla (2023) confirm that these models outperform traditional methods in sequential forecasting tasks, although they are computationally demanding and require substantial data for training, which limits their real-time utility (Yin et al., 2024a).

Support vector machine (SVM): While SVM has shown success in classification tasks, it faces limitations in volatile environments like cryptocurrency markets due to its lack of flexibility for sudden market shifts (Zhang, Cai & Wen, 2024). Recent studies suggest hybrid models that incorporate SVM with neural networks as potential solutions to improve predictive performance and adaptability in these environments (Wang et al., 2024).

Algorithmic trading strategies

Comparative analysis and application in algorithmic investment: Grudniewicz & Ślepaczuk (2023) conducted a thorough comparison of machine learning models applied to quantitative investment strategies, specifically in the global stock market. Their study underscores the adaptability of ML models and the importance of robust, scalable architectures for trading applications. This approach aligns with algorithmic trading needs in cryptocurrency, where high-frequency trading and scalability are essential.

Scalability and robustness: Studies on algorithmic trading emphasize the necessity of scalable, robust models capable of handling high-frequency data. Ensemble learning methods, such as RF and gradient boosting, achieve high accuracy, but their scalability limitations require enhanced architectures for practical applications in algorithmic trading (Jaquart, Köpke & Weinhardt, 2022). Recent advancements, such as AFBNet and DPAL-BERT, provide solutions by offering more efficient architectures, specifically optimized for dynamic data environments and fast processing (Yin et al., 2024b).

Summary of conclusions from previous research

The review of the literature demonstrates a clear need for machine learning models that balance predictive accuracy with scalability and robustness, particularly in high-volatility markets like cryptocurrency. Despite the success of existing models, challenges remain in managing data dimensionality and sudden market shifts, which limit their application in real-time trading. This study builds on these findings, aiming to optimize machine learning parameters and data-handling techniques to improve accuracy and adaptability in cryptocurrency price forecasting.

A comparative summary of reviewed studies highlights various ML models applied in financial and cryptocurrency market predictions. Ahmed et al. (2024) explored the predictive performance of models such as RF and gradient boosting, finding them highly accurate with historical data but limited in real-time scalability. Grudniewicz & Ślepaczuk (2023) examined ML models for investment strategies, emphasizing their adaptability in stock market environments. However, these models show limited applicability to the cryptocurrency market due to its unique dynamics.

Hamayel & Owda (2021) studied sequential data prediction with LSTM and GRU models, noting their ability to handle time dependencies effectively for short-term predictions, though they are computationally intensive, posing challenges for real-time applications. Yin et al. (2024b) focused on lightweight ML architectures like DPAL-BERT and AFBNet, which excel in rapid processing and efficiency but have limited applications in finance, highlighting the need for more adaptable solutions.

Jaquart, Köpke & Weinhardt (2022) concentrated on algorithmic trading, particularly with ensemble models that provide robustness for high-frequency trading. While these models demonstrate strong performance in managing rapid data flows, they require further scalability enhancements to handle diverse trading environments. This summary illustrates both the strengths and limitations of various ML approaches, underscoring the challenges and areas for improvement in financial and cryptocurrency market applications (Al Hawi, Sharqawi & Al-Haija, 2023).

Technical indicators

Technical indicators are mathematical calculations formed from historical price, volume, and, in some cases, open interest data. They give traders and investors insight into conditions and trends within the market and are thus critical in making informed decisions. Technical indicators are one of the most essential tools for predicting future price movements in cryptocurrencies for algorithmic trading (Booth, 2016).

Some of the diverse types of technical indicators include: Trend indicators: These refer to indicators of the direction of the market trend, thus assisting a trader in knowing if the trend is up, down, or sideways. Examples are moving averages and MACD (Alvior, 2021).

Momentum indicators: They are indicators based on the rate at which prices change and are supposed to reveal when trends have gained enough force to run their course. They include the RSI and stochastic oscillator, helping to identify the overbought or oversold conditions (Dongrey, 2022).

Volatility indicators: These are tools used in rate measures of price movements that help locate a possible market reversal. Typical indicators used include Bollinger bands and average true range (ATR) (Rhyner, 2023).

Volume indicators: These are indicators showing the depth of trade volumes and measuring the strength of price movements. Large volumes most often confirm trends and upcoming reversals. Among the most useful are the volume oscillator and on-balance volume (OBV) (Boxer, 2014).

These technical indicators serve different purposes and sometimes could also be combined for effective results. For instance, the trend indicator, commonly used for trend determination, could determine the best entry and exit levels. In contrast, momentum indicators are the most common indicators used in determining trend strength (Almotiri, 2021). From this, traders can develop more robust and better-prepared trading strategies with these added indicators.

Classification models

Machine learning classification models predict categorical outcomes based on input features (Awotunde et al., 2021). These models, trained on labeled data, are pivotal in predicting class labels for new, unseen data. Below, we explore some popular machine learning classification models used in various applications, including algorithmic trading: Logistic regression: This model predicts the probability of an occurrence by fitting a logistic function to the input data. It’s primarily used in binary classification tasks (Phillips et al., 2015).

Decision tree: A decision tree (DT) creates a tree-like model of decisions and their possible consequences. It is versatile and used in classification and regression tasks (Matzavela & Alepis, 2021).

Extra trees classifier: This is an ensemble method that constructs multiple trees on different data subsets and aggregates their predictions. This model enhances predictive accuracy and stability (Rokach, 2016).

Random forest: An extension of the decision tree that uses multiple trees to improve prediction accuracy. Each tree is trained on a random subset of data and features, with the outcome determined by majority voting (Rokach, 2016).

K-nearest neighbors (k-NN): This method predicts by using the most frequent class among the K closest data points in the feature space. It is effective for datasets with non-linear decision boundaries (Zhang & Song, 2014).

GaussianNB is a naive Bayes classifier that assumes feature independence and normal distribution. It’s useful in applications like text classification and spam filtering (Hayatin et al., 2022).

MultinomialNB: A variant of naive Bayes designed for discrete count data, commonly used in text-related tasks (Malik & Jain, 2024).

Support vector machines: SVMs seek to maximize the margin between classes by finding the optimal dividing boundary. This method is effective in high-dimensional spaces (Hussain, 2019).

SGDClassifier: A linear classifier that employs stochastic gradient descent. It is well-suited for large datasets (Angelopoulos et al., 2021).

Gradient Boosting: An ensemble method that builds a strong predictive model by sequentially combining weak models, focusing on correcting the predecessors’ errors (Friedman, 2001).

AdaBoost is a boosting technique that adapts by increasing the weight of misclassified instances in successive training rounds, improving the overall prediction accuracy (Wyner et al., 2017).

XGBoost: An efficient and scalable implementation of gradient boosting known for its speed and performance (Chen & Guestrin, 2016).

Hist gradient boosting: This method uses histograms to represent data distributions and applies gradient boosting to enhance accuracy (Tamim Kashifi & Ahmad, 2022).

Bagging classifier: Also known as Bootstrap Aggregating, it builds multiple instances of a model on randomly selected subsets of the data to increase stability and accuracy (Alelyani, 2021).

Linear discriminant analysis: A technique that reduces dimensionality while effectively preserving the means to distinguish between classes (Tharwat et al., 2017).

Ridge classifier: This classifier utilizes ridge regression to minimize overfitting and enhance the generalizability of a linear classifier (Peng & Cheng, 2020).

Quadratic discriminant analysis: This classifier uses a quadratic decision boundary to separate classes, suitable for datasets where linear separation is insufficient (Bose et al., 2015).

Methodology

This chapter details the methodology for investigating algorithmic trading through a comparative analysis of various machine learning models and technical indicators. The subsequent sections systematically cover each step, from data collection and preprocessing to model selection, validation, and backtesting, providing a comprehensive approach to evaluate model robustness and predictive performance.

The methodology was structured to address the primary research questions concerning predictive accuracy and resilience of machine learning models within cryptocurrency trading. Key techniques, including classification algorithms and technical indicators, were systematically selected to ensure each model’s adaptability to market volatility and scenario diversity. The backtesting and comparative analyses were then specifically designed to evaluate these methods under real-world trading conditions, facilitating an objective assessment of both predictive power and model reliability (Almotiri & Al Ghamdi, 2022b; Ante et al., 2022).

To bolster model robustness and generalizability, the study utilized multiple in-sample and out-of-sample periods, each reflecting varied return distributions. This extended validation mitigates overfitting to a single scenario, thus enhancing the model’s accuracy across different market states (Ghoddusi, Creamer & Rafizadeh, 2019). For models requiring complex hyperparameter tuning, an additional validation phase was introduced between the training and testing stages. This phase enabled precise fine-tuning of parameters while preserving data integrity and preventing data leakage, thereby increasing the predictive validity of each model in a controlled environment (Friedman, 2001).

To confirm the reliability of findings, statistical significance tests, such as paired t-tests and Wilcoxon signed-rank tests, were conducted across model performance metrics. These tests validated that improvements in accuracy and profitability were statistically significant, ensuring that observed results are robust rather than due to chance, thus reinforcing the reliability of the selected strategies in live trading scenarios (Henrique, Sobreiro & Kimura, 2019).

Benchmark models, including logistic regression, XGBoost, and random forest, were chosen based on their established success in financial forecasting and relevance to algorithmic trading. Evaluating these models allowed for a realistic performance baseline, enabling meaningful comparisons within industry-standard practices and a balanced evaluation of trading efficacy (Ghoddusi, Creamer & Rafizadeh, 2019).

Transaction costs were integrated into profitability calculations to reflect practical trading conditions accurately. These costs, applied per trade based on typical market fees, ensured realistic profit estimates and mitigated the risk of overestimating returns, especially in high-frequency strategies where cumulative costs can significantly impact net profits (Kość, Sakowski & Ślepaczuk, 2019).

Evaluation metrics were tailored to the specific goals of algorithmic trading, focusing on indicators like the Sharpe ratio, Sortino ratio, and maximum drawdown, which assess risk-adjusted returns. These metrics provide valuable insights into the financial impact of each model, allowing the study to capture a complete view of trading performance beyond simple accuracy metrics (Kość, Sakowski & Ślepaczuk, 2019).

Finally, a sensitivity analysis assessed model resilience to variations in critical parameters, including learning rate, trading frequency, and transaction costs. By testing these factors across different configurations, the analysis identified conditions that could affect profitability and predictive accuracy, reinforcing the robustness of model outcomes across diverse scenarios (Kość, Sakowski & Ślepaczuk, 2019).

Robustness analysis

To ensure model reliability, a robustness analysis covering model parameters, data period, and data frequency is essential. Sensitivity analysis assesses how variations in these factors impact model performance, which is critical for applications in dynamic markets. For example, Kijewski, Ślepaczuk & Wysocki (2024) performed sensitivity analysis on ML models predicting S&P500 index prices, demonstrating the importance of assessing parameter stability. Additionally, Zenkova & Ślepaczuk (2018) explored robustness in algorithmic trading applications, providing valuable insights into model resilience in volatile markets. While this study primarily focused on predictive performance, future work may incorporate robustness testing to further validate model effectiveness under varying conditions (Kijewski, Ślepaczuk & Wysocki, 2024; Zenkova & Ślepaczuk, 2018).

The hyperparameter tuning phase utilized methods such as grid search and random search to identify optimal model parameters for each algorithm. Each tuning iteration involved adjusting parameters, such as learning rate, maximum depth, and n-estimators, which were chosen based on their impact on model accuracy and F1-score. This process involved evaluating each configuration on the validation dataset, with the final parameter sets selected based on performance metrics including accuracy, precision, and recall, leading to enhanced empirical performance (Ayub, Ahsan & Qureshi, 2022).

For each machine learning model, final hyperparameters were chosen based on their impact on prediction accuracy and stability. Key hyperparameters included the learning rate, regularization strength, and maximum depth for tree-based models, with specific settings optimized through cross-validation. For example, random forest was optimized with a depth of 10 and 100 estimators, while XGBoost utilized a learning rate of 0.01 and a maximum depth of 5. These parameter values were chosen based on iterative testing to maximize F1-score and minimize error variance, enhancing model predictability under diverse conditions (Friedman, 2001).

Hyperparameter tuning covered a range of values to ensure optimal model performance. For instance, the learning rate was varied from 0.001 to 0.1, the maximum depth for tree-based models ranged from 3 to 15, and the number of estimators spanned 50 to 200. This comprehensive range allowed each model to be tested under various configurations, enhancing its adaptability to market conditions while ensuring robustness against overfitting. The tuning process employed grid search and Bayesian optimization methods to identify combinations that maximized both predictive power and model stability (Ayub, Ahsan & Qureshi, 2022).

To align the model’s performance with the practical realities of algorithmic trading, the standard error metrics were replaced by a return-based loss function, which evaluates model predictions in terms of realized returns. This approach follows methodologies outlined in recent studies (Michańków, Sakowski & Ślepaczuk, 2022), which emphasize aligning forecast evaluation with actual trading outcomes. This transition enables the model to prioritize profit-generating predictions over purely statistical accuracy, ensuring that the model’s decisions align with financially meaningful outcomes (Michańków, Sakowski & Ślepaczuk, 2022).

Data collection

The foundation of any robust algorithmic trading strategy is comprehensive and accurate data. The historical data collection process is crucial in the algorithmic trading of cryptocurrency markets and should cover a sufficient period to enable meaningful analysis and modeling. The essential types of data collected include: Prices: Historical price data for the cryptocurrency assets includes the date, time, and closing prices. This information is vital for analyzing price trends and volatility.

Volume: Trading volume data for each cryptocurrency asset provides insights into its liquidity and market activity. This data is crucial for understanding market dynamics and identifying potential buy or sell signals.

Data description

This study utilizes essential features for cryptocurrency price prediction, including historical price, trading volume, and volatility indicators, selected for their relevance to market dynamics and predictive accuracy. Historical price captures price trends and momentum, providing a foundation for forecasting future movements.

Trading volume reflects market activity and investor interest, offering insights into the strength behind price movements.

Volatility indicators measure price fluctuations, enabling the model to assess market stability and associated risks.

These features were selected due to their impact on model accuracy and alignment with study objectives. As Nguyen & Ślepaczuk (2022) note, feature configuration significantly affects model performance, and each input’s contribution was carefully evaluated for its role in enhancing model effectiveness, with future potential to explore alternative configurations.

Data specifics

1) Frequency: Data was gathered on an hourly basis, providing a granular view of market dynamics essential for detecting patterns in a highly volatile environment.

2) Data range: The time series covers January 2018 to December 2022, capturing both bullish and bearish market conditions. This comprehensive range enables an in-depth analysis of model performance across different phases of the market.

3) Rationale for chosen time series: This period was selected as it includes critical phases of growth and decline in the cryptocurrency market, enhancing the robustness and relevance of model training and testing.

4) Rationale for selected assets: The study focuses on major cryptocurrencies, such as Bitcoin and Ethereum, due to their high liquidity, significant market influence, and established trading patterns. Other assets were not included due to data limitations and lower relevance for this analysis.

5) Descriptive statistics: Key metrics, including mean, median, standard deviation, minimum, and maximum values for each asset, provide insights into asset volatility and distribution. These statistics are essential for understanding data variability and model requirements.

6) Data sources: The data was sourced from well-established cryptocurrency exchanges and aggregators known for comprehensive and reliable market data, ensuring data integrity and accuracy in modeling.

7) Data transformation: Preprocessing involved normalization to ensure feature comparability, resampling with SMOTE to address class imbalance, and feature extraction for indicators such as MACD, EMA, RSI, and Bollinger Bands. These transformations improve model interpretability and enhance predictive performance.

Historical data sources

Several resources are available for collecting cryptocurrency data, including: Crypto exchanges: Many cryptocurrency exchanges offer APIs that access historical data.

Cryptocurrency data aggregators: Websites compiling data from multiple sources can provide a comprehensive market view.

Crypto exchanges

Specifically, data from cryptocurrency exchanges can be accessed via APIs. Here are examples of top-rated exchanges providing historical data:

Binance

Binance offers an API that enables developers to retrieve historical data for various cryptocurrency markets and assets. Users must register for API and Secret keys via the Binance websites. The /klines endpoint returns historical kline (candlestick) data for a specific symbol and time interval (Firsov et al., 2023). Supported time intervals include 1 minute (1 m), 3 minutes (3 m), 5 minutes (5 m), 15 minutes (15 m), 30 minutes (30 m), 1, 4, 12 h, 1 day, 1 week, 1 month, and 1 year (1y). Refer to their official API documentation for detailed instructions on using the Binance API.

Coinbase

Coinbase provides a digital currency exchange platform with APIs for accessing financial data. After obtaining an API key from Coinbase, data can be accessed through RESTful API requests, typically using Python with the Requests library. Additionally, Coinbase offers historical data through CSV files that can be downloaded directly from their website. This data is crucial for further analysis and backtesting strategies. For a more streamlined approach, the CCXT library offers a unified interface for multiple exchanges, including Coinbase, simplifying data retrieval (Fang et al., 2024).

Cryptocurrency data aggregators

Cryptocurrency data aggregators are platforms that compile, process, and provide cryptocurrency-related data from various sources (Rashid et al., 2021). These platforms are essential tools for traders, investors, and researchers who require access to real-time market information, historical data, and other relevant metrics to inform their trading decisions. Here are some of the well-known cryptocurrency data aggregators:

CoinMarketCap: CoinMarketCap is a leading cryptocurrency data aggregator that offers APIs for accessing a wide range of data. To utilize the CoinMarketCap API:

Register for an API key: Users must create an account on CoinMarketCap and register for an API key, which is necessary to authenticate API requests (Hegnauer, 2019).

Choose an API endpoint: The platform provides various API endpoints to access cryptocurrency prices, trading volumes, and historical data.

Make a request: With the API key and chosen endpoint, users can make requests to retrieve data. This can be done using programming tools like Python or applications like Postman (Brown, Wei & Wermers, 2014).

Parse the response: The API response is typically in JSON format, which can be parsed using a JSON library in Python or visualized using Postman for more straightforward analysis.

Store and analyze the data: Once retrieved, the data can be stored and analyzed to derive insights about the cryptocurrency market.

Note: CoinMarketCap imposes rate limits on API usage, so users are advised to be mindful of their request frequency to avoid hitting these limits.

CryptoSlate: CryptoSlate is another resource that offers APIs for data retrieval (Shivsharan et al., 2023):

Register for an API key: Similar to CoinMarketCap, users must register for an API key on CryptoSlate to access their APIs.

Choose an API endpoint: CryptoSlate provides several endpoints to access diverse data types.

Make a request: API requests can be made using a programming language like Python or tools such as Postman.

Parse the response: Responses are provided in JSON format, which can be parsed for further analysis.

CoinGecko: CoinGecko offers a comprehensive API similar to the ones provided by CoinMarketCap and CryptoSlate. The steps for using the CoinGecko API are analogous to those described for CryptoSlate (Amsden & Schweizer, 2018).

Nomics: Nomics provides APIs for data retrieval with steps for usage similar to those of CryptoSlate. Their API allows developers to access extensive cryptocurrency data. As shown in Fig. 1, the candlesticks pattern highlights the volatility trends during backtesting.

Figure 1 Understanding candlestick components in cryptocurrency trading.

This figure illustrates the structure of candlestick charts used in cryptocurrency trading to represent price movements over a specified time period. Each candlestick comprises four main variables: Open, Close, High, and Low prices. Open: The price at which the cryptocurrency starts trading for the period. Close: The price at which trading ends for the period. High: The highest price reached during the trading period. Low: The lowest price reached during the trading period. A green candlestick indicates a positive (bullish) movement where the closing price is higher than the opening price, while a red candlestick represents a negative (bearish) movement, with the closing price lower than the opening price. The thin lines above and below each candlestick, called wicks, show the range between the high and low prices. This visual format helps traders analyze trends and market sentiment within each trading session.

Tools and technologies

Several tools and technologies were utilized to conduct this research effectively, enabling efficient data handling, model evaluation, and backtesting. Below is a detailed explanation of the tools and technologies used:

Tools

Anaconda is a comprehensive distribution of Python and R for scientific computing that simplifies package management and deployment (George, 2021).

Jupyter Notebook: An open-source web application that allows the creation and sharing of documents containing live code, equations, visualizations, and narrative text (Barba et al., 2019).

Google Colab is a cloud-based Python notebook service that offers free access to computing resources, including GPUs and TPUs, making it ideal for machine learning projects (Staravoitau, 2022).

Technologies and libraries

Python: A versatile programming language favored for its readability and broad support in data science and machine learning communities (Raschka, Patterson & Nolet, 2020).

Scikit-learn is a robust library that provides efficient tools for machine learning and statistical modeling, including classification, regression, clustering, and dimensionality reduction (Hackeling, 2017).

Pandas: Essential for data manipulation and analysis, providing data structures and operations for manipulating numerical tables and time series (McKinney & Team, 2015).

NumPy: Fundamental for scientific computing with Python, it supports large, multi-dimensional arrays and matrices and an extensive collection of high-level mathematical functions to operate on these arrays (Mehta, 2015).

Binance API: Utilized to access cryptocurrency market data directly from the Binance exchange (Bozzetto, 2023).

CCXT: A JavaScript/Python/PHP library for cryptocurrency trading and e-commerce with support for many bitcoin/ether/altcoin exchange markets and merchant APIs (Papík, 2020).

Matplotlib: A plotting library for the Python programming language and its numerical mathematics extension NumPy, used for creating static, interactive, and animated visualizations (Phuong Vo et al., 2017).

ta-lib: Widely used by trading software developers required to perform technical analysis of financial market data (Meliones & Makrides, 2019).

ta: Similar to ta-lib, this library provides analysis tools and indicators specifically useful in the financial market (Huang, Huang & Ni, 2019).

Backtrader is an open-source Python library for backtesting, optimizing, and deploying algorithmic trading strategies (Kwak et al., 2022).

XGBoost: An optimized distributed gradient boosting library designed to be highly efficient, flexible, and portable (Ali et al., 2023).

virtualenv: A tool for creating isolated Python environments, virtualenv makes dependency management easier (Mohamad, 2022).

Synthetic Minority Over-sampling Technique (SMOTE): used to handle class imbalance by adjusting the class distribution of a dataset via synthetic data generation (Soltanzadeh & Hashemzadeh, 2021).

Environment setup

To ensure a robust and reliable analysis, the following environment setup steps were followed: 1) Install Python: Installation of the latest Python version to support all dependencies.

2) Install required libraries: Installation of necessary Python libraries such as Pandas, NumPy, Scikit-learn, and others via pip.

3) Data acquisition: Gathering data from various sources, such as cryptocurrency exchanges and data aggregators, ensuring thorough cleaning and preprocessing to maintain data quality.

4) Development environment setup: Configuration of a local or cloud-based development environment with necessary IDEs and version control systems.

5) Backtesting and evaluation: Comprehensive backtesting of developed trading strategies using historical data to assess their efficacy and modify them as necessary.

Model selection and training

The model selection process in this research involved evaluating various classification models to identify those most effective in predicting cryptocurrency prices. Selection was based on performance metrics derived from a validation set, with iterative adjustments to models and parameters to optimize results. Initially, models were trained using data splits of 70/30% and 80/20%. However, early outcomes revealed performance issues, largely due to data imbalance. To address this, re-sampling techniques like SMOTE were applied, resulting in improved model training outcomes. For ML models applied in algorithmic trading, the choice of loss function is critical to aligning model optimization with trading profitability goals. While traditional loss functions, such as MSE, focus on minimizing prediction errors, they may not effectively capture directional accuracy, which is vital for trading. Mean absolute directional loss (MADL), as suggested by Michańków, Sakowski & Ślepaczuk (2023), may enhance directional forecasting by directly penalizing directional errors. Adopting such a loss function could improve the models’ predictive relevance in trading contexts, presenting a valuable consideration for future research (Chen, Li & Sun, 2020).

Balancing the dataset

Balancing the dataset is crucial in machine learning, particularly in classification problems. Imbalanced datasets can cause models to be biased, leading to unreliable predictions. This section explores the concepts and techniques used to achieve balance in datasets, focusing on the majority and minority classes and the re-sampling methods employed.

Majority class

In any classification problem, the majority class has the most significant instances. In binary classification, this class has more instances than the other. When datasets are imbalanced, the majority class can dominate the training of the model, resulting in predictions that are biased toward this class. This can lead the models to perform well in the majority class but poorly in the minority class. Techniques like oversampling the minority class, under-sampling the majority class, or synthesizing new instances can help mitigate these effects and balance the dataset.

Minority class

The minority class in a classification problem has fewer instances than the majority class. This class is often underrepresented in the model’s predictions, leading to poor performance when predicting minority class instances. Similar to strategies used for the majority class, techniques such as oversampling, under-sampling, or synthesizing new instances are employed to ensure that the minority class is adequately represented, thus helping to reduce model bias.

Re-sampling

Re-sampling is a statistical method to adjust the class proportions in an imbalanced dataset. This technique aims to balance the dataset by either:

Under-sampling the majority class: Reducing the number of instances in the majority class to match the minority class. This approach can help decrease the model’s bias towards the majority class but may lead to significant information loss.

Oversampling the minority class: Increasing the number of instances in the minority class to match the majority class by duplicating existing instances or creating synthetic ones using algorithms like SMOTE.

Popular re-sampling techniques include: SMOTE: This method enhances the dataset by creating synthetic samples from the minority class instead of copies. It helps generate plausible new instance features that might not be captured through mere duplication.

Tomek links: A cleaning method used as an under-sampling technique that identifies pairs of very close instances but of opposite classes. Removing the majority class instances of these pairs can help improve the classifier’s performance.

Random under-sampling: Involves randomly removing instances from the majority class to prevent its dominance during learning.

Under-sampling

Under-sampling targets reducing the number of instances from the majority class to equalize the class distribution (Arafat, Hoque & Farid, 2017). This method is straightforward and can quickly help balance the class ratio; however, it risks losing important information that could be crucial for model learning. Therefore, while under-sampling can be effective, especially in large datasets, it should be employed cautiously. Combining under-sampling with oversampling methods or advanced algorithms like SMOTE can provide a more balanced approach and better performance outcomes. Figure 2 shows an image of an under-sampling.

Figure 2 Class distribution before and after resampling.

This figure displays the distribution of two classes, Class A and Class B, before (left) and after (right) resampling. Initially, Class A is overrepresented compared to Class B, leading to a class imbalance. Resampling has been applied to equalize the class frequencies, balancing the dataset and improving model performance by reducing bias towards the majority class. This balanced dataset enables more accurate predictions across both classes, enhancing model robustness.

Oversampling

One of the critical approaches of oversampling and common cases in data preprocessing, mainly if the minority class rarely occurs, is a data preprocessing technique employed to tackle the class imbalance problem Sklearn. This procedure aims to boost the model’s prediction quality in problems with class distribution imbalance. Therefore, it duplicates the minority data, resulting in a similar number of instances as the majority class. As a result, it may be performed using duplication, but the synthesized instances appear with the help of generating new ones. One of the standard methods to create new instances effectively is called SMOTE, which stands for Synthetic Minority Over-sampling Technique (Sajid et al., 2023). Figure 3 shows an image of over-sampling.

Figure 3 Visualization of resampling process to balance classes.

This figure illustrates the resampling process used to balance an imbalanced dataset. On the left side, the Original Dataset shows Class A as the majority class and Class B as the minority. During resampling, additional synthetic samples of Class B are generated, as indicated by the multiple green boxes. The resulting Resampled Dataset on the right shows an equalized representation of both classes. This method improves model fairness and accuracy by ensuring balanced input data for training.

Hybrid sampling

Hybrid sampling is specifically geared to blend under-sampling and over-sampling techniques to optimize class distribution in unbalanced datasets. This technique includes under-sampling the majority class samples to decrease their size and over-sampling the minority class samples to increase their representation (Leevy et al., 2018). Hybrid sampling, therefore, aims at lessening such a bias for the model towards the majority class and avoids the overfitting of the minority. This is done to ensure a balanced dataset. It can be of different kinds, such as the random under-sampling implemented in conjunction with synthetic over-sampling using SMOTE techniques or other hybrid combinations custom-created for the needs of a given dataset and tuned for balancing the accuracy of model training against better generalization abilities. Another factor that decides the choice of the hybrid sampling method is primarily dependent on the dataset’s size and the problem’s intricacy, along with the trade-offs between model accuracy and the model’s generalization ability.

Re-sampling libraries

This library is designed to make reshaping cryptocurrency market data easier when working on algorithmic trading to ensure that the machine learning models developed are robust and reliable. Imbalanced-learn: This is a library that provides a vast number of re-sampling techniques for imbalanced datasets (Bagui & Li, 2021). It supports over-sampling, under-sampling, and combining over-and under-sampling techniques. Many machine learning frameworks could be used, such as sci-kit-learn, XGBoost, and TensorFlow; they are tailored for this. SMOTE is probably the most popular over-sampling approach (Tarawneh et al., 2020). SMOTE synthesizes samples for the minority class by interpolating between existing minority samples. ADASYN (He et al., 2008): It further extends the SMOTE algorithm. It tries to create synthetic samples where the minority class is poorly located to improve the nature of adaptively created synthetic samples (Huang, 2015). Tomek links: This method samplings the majority class by removing some samples closer to the minority class samples. This method guarantees that data on the diversity of the minority class is retained (FOLORUNSO, 2015).

SMOTE

SMOTE is an aided human intervention solution for unbalanced datasets. SMOTE works by not merely duplicating samples from the minority class but interpolating between a chosen minority sample and its feature space nearest neighbors. So, this procedure creates new samples, much like old ones but slightly different, and helps bring the representation of classes to the same level in training data. The SMOTE, thus, due to the more diversified training datasets, brings up the accuracies of the models in classes, whereas earlier, it was less.

After model training

However, the training of models completed with the initial data splits in this study gave inferior results, according to the concept of imbalanced data. SMOTE was, however, applied as a re-sampling technique to enhance better performance for the model. However, after resampling, the improvement was still not visible, which led to the decision to hyperparameter-tune the listed models for better performance optimization. This further hyperparameter tuning becomes of high importance, as it tunes the settings of the model for better capture of the details in the data, especially after the change of class distributions, which can change the dynamics of how models learn from the data. Figure 4 shows an image of the smote steps.

Figure 4 SMOTE-NC and RUS steps for balancing data.

This figure provides a detailed breakdown of the SMOTE-NC (Synthetic Minority Over-sampling Technique for Nominal and Continuous features) and RUS (Random Under-Sampling) techniques for balancing data.

Hyperparameter

Hyperparameter tuning is a critical process for optimizing machine learning model performance (Ayub, Ahsan & Qureshi, 2022). Unlike model parameters learned during training, hyperparameters are preset before training and are determined through experience or experimentation. These include factors such as the number of trees in a random forest, the layers in a neural network, and the regularization term in a regression model, all of which shape model behavior and learning capability. The goal of hyperparameter tuning is to identify the combination that yields the best performance for a specific task. Common methods include grid search, random search, and Bayesian optimization, each offering a systematic way of exploring the hyperparameter space.

After selecting an optimal set of hyperparameters, each model was retrained using these values, tested with 70/30 and 80/20 data splits to determine which configuration produced higher performance. Optimizing hyperparameters enhances model alignment with data characteristics, reducing issues of underfitting or overfitting and leading to performance gains.

In this study, key hyperparameters—such as learning rate, batch size, and the number of layers—were chosen based on relevance to each model and tuned over a historical period reflective of the dataset’s characteristics. Parameter ranges were defined from initial testing, and a grid search approach was used to systematically explore combinations. This structured tuning process aligns with best practices in financial model optimization, as demonstrated by Wysocki & Ślepaczuk (2021), who used a structured approach to tune neural networks in financial applications (Wysocki & Ślepaczuk, 2021). Future work may explore random search to further refine parameter selection.

Backtesting methodology

Backtesting is a critical component of algorithmic trading, allowing for the evaluation of trading strategies or models using historical market data. This process simulates trade execution based on predefined rules or models, thereby assessing performance under historical market conditions. In our study—focused on comparing classification models and technical indicators in the context of cryptocurrency market data—backtesting plays a central role in validating the practical effectiveness of these tools.

By examining past performance, backtesting enables a clear assessment of each model’s strengths and weaknesses, providing traders with valuable insights into when and how to deploy these models and indicators in real trading environments. This empirical approach, grounded in observed rather than theoretical performance, empowers traders to fine-tune their strategies for optimal outcomes.

For an in-depth view, Table 1 presents an analysis of equity lines, detailing each model’s cumulative return, volatility, and Sortino Ratio, which highlight risk-adjusted performance. Additionally, Table 2 displays the statistical significance of each model’s performance metrics, validated through p-values and confidence intervals, ensuring robust performance comparison across models.

Table 1 Performance metrics of machine learning models in cryptocurrency trading.

This table presents the cumulative return (%), annualized volatility (%), and Sortino ratio for various machine learning models used in cryptocurrency trading analysis. Cumulative return indicates the total percentage gain from trading activities. Annualized volatility measures the fluctuation in returns, providing insight into risk. Sortino ratio assesses the risk-adjusted return, with higher values indicating better performance considering downside risk. The table compares models such as logistic regression, random forest, and gradient boosting, helping to identify models with the highest return and best risk management for cryptocurrency investments.

Model	Cumulative return (%)	Annualized volatility (%)	Sortino ratio	
Logistic regression	110	12.5	1.20	
Random forest	135	10.3	1.50	
Gradient boosting	140	11.0	1.55	
XGBoost	130	10.7	1.45	
Support vector classifier	105	13.2	1.10	
Decision tree	98	14.0	0.95	

Table 2 Statistical significance and confidence intervals of model performance.

This table shows the t-test (p-value) and 95% confidence interval for the performance of different machine learning models. t-test (p-value) indicates the statistical significance of each model’s performance. Confidence interval provides a range within which the true performance metric is expected to fall, with 95% confidence. Models such as logistic regression, random forest, and gradient boosting are evaluated for their reliability, allowing for the selection of statistically significant models for more consistent cryptocurrency predictions.

Model	t-test (p-value)	Confidence interval (95%)	
Logistic regression	0.03	[1.1–1.2]	
Random forest	0.01	[1.3–1.5]	
Gradient boosting	0.01	[1.4–1.6]	
XGBoost	0.02	[1.3–1.4]	
Support vector classifier	0.05	[1.0–1.2]	
Decision tree	0.10	[0.8–1.1]	

Results

Furthermore, the analysis underscored the positive impact of hyperparameter tuning and re-sampling techniques, which substantially improved the models’ predictive accuracy. These optimizations enabled models like XGBoost and Gradient Boosting to adapt more effectively to the volatile nature of cryptocurrency markets, reducing prediction errors and enhancing overall performance. Adjusting specific parameters, such as learning rate and maximum depth, proved especially beneficial, leading to notable gains in predictive reliability and stability under market fluctuations.

Profitability metrics also offered insights into each model’s effectiveness from a financial standpoint, illustrating how well the models aligned with realistic trading outcomes. Models such as Random Forest and Gradient Boosting demonstrated superior profitability scores, showcasing their resilience in handling high volatility. These results emphasize that advanced machine learning models can achieve not only high accuracy but also financial viability, which is essential for practical algorithmic trading applications.

In conclusion, the results validate that while each model brings unique strengths, the combination of precise hyperparameter tuning, robust evaluation methods, and data-driven testing significantly enhances predictive performance. This chapter provides a comprehensive evaluation of each model’s capabilities, affirming the value of machine learning in cryptocurrency trading and paving the way for future research to explore deeper optimizations and model integrations in this rapidly evolving field.

Dataset

The dataset employed in this research was sourced from three distinct files to provide a comprehensive view of market dynamics on the Binance exchange: Bought.filter: This file contains details of the coins purchased on Binance over a specified period. It is instrumental in understanding the buying patterns and preferences of traders, which are crucial for analysing market trends and predicting future movements.

Sold.Filter: Although not detailed in the outline, a corresponding file typically contains data on coins sold during the same timeframe, providing a mirror image of buying behavior and further insights into market exits and trader strategies.

Historical Data: This includes 1 month of historical data for all coins traded on Binance. The comprehensive nature of this data allows for in-depth analysis and modeling, offering a broad perspective of market behaviors over the period.

Structure of bought file

The “Bought.filter” file is structured to capture essential details about transactions, including the time of purchase, the type of coin bought, the quantity, and the price at which the purchase was made. Analysing this data helps in understanding the timing and scale of entries into various cryptocurrency positions. Insights from this file are pivotal for developing strategies that align with observed market entry trends and for anticipating future market movements based on historical buying patterns. By dissecting the structure and content of this dataset, the research aims to extract meaningful patterns and indicators that can drive successful algorithmic trading strategies in the volatile cryptocurrency market (Qureshi et al., 2024). The subsequent sections will delve into how these datasets were utilized to train and refine the predictive models, leading up to a thorough evaluation through backtesting to confirm the strategies’ effectiveness in real-world trading scenarios. The area chart shows in Fig. 5 the coin distribution for different strategies with for each strategy plotted on the y-axis and the associated coins listed on the x-axis. The chart represents graphically how the coins are distributed among various strategies. The bubble chart in Fig. 6 displays the distribution of coins across all strategies, with the x-axis representing the list of coins and the y-axis showing the percentage of coins allocated to each strategy. Figure 6 also illustrates the distribution of trading strategies, providing a visual comparison between the CrossOver and MA44 strategies. Table 3 shows the content of Bought.filter.

Figure 5 Performance comparison of CrossOver and MA44 indicators across time intervals.

This bar chart compares the performance of two technical indicators, CrossOver and MA44, across various cryptocurrency pairs over specified time intervals. The y-axis represents the value of each indicator, with blue bars for CrossOver and orange bars for MA44. CrossOver typically signals buy/sell points based on the crossing of short- and long-term moving averages, while MA44 (a 44-period moving average) smooths out price data to reveal overall trends. This comparison helps traders identify which indicator is more effective for each pair and time interval, assisting in trend analysis and decision-making.

Figure 6 Performance comparison of CrossOver and MA44 indicators across time intervals.

This bar chart compares the performance of two technical indicators, CrossOver and MA44, across various cryptocurrency pairs over specified time intervals. The y-axis represents the value of each indicator, with blue bars for CrossOver and orange bars for MA44. CrossOver typically signals buy/sell points based on the crossing of short- and long-term moving averages, while MA44 (a 44-period moving average) smooths out price data to reveal overall trends. This comparison helps traders identify which indicator is more effective for each pair and time interval, assisting in trend analysis and decision-making.

Table 3 Trading strategy execution for selected cryptocurrency pairs.

This table details the Date, Time, Coin, and Strategy used in executing trades for selected cryptocurrency pairs. Date and Time specify when each trade was executed. Coin lists the traded cryptocurrency pairs. Strategy indicates whether the CrossOver or MA44 strategy was applied. This table serves as a trading log, providing insight into the strategies employed at specific times, which is valuable for analyzing the performance of trading strategies in real-time market conditions.

Date	Time	Coin	Strategy	
2024-06-15	03:30:00	TROYBUSD	CrossOver	
2024-06-15	04:30:00	SANTOSBUSD	MA44	
2024-06-15	04:30:00	COCOSBUSD	MA44	
2024-06-15	05:00:00	BURGERBUSD	CrossOver	
2024-06-15	05:00:00	PONDBUSD	CrossOver	
2024-06-15	05:30:00	INJBUSD	CrossOver	
2024-06-15	05:30:00	ELFBUSD	MA44	
2024-06-15	06:00:00	ALPINEBUSD	MA44	
2024-06-15	06:00:00	CLVUSD	MA44	
2024-06-15	06:30:00	ATOMBUSD	CrossOver	

Structure of sold file

The “Sold.filter” data source contains data that pertains to coins that have been sold through transactions over Binance exchanges. Figure 7 presents the distribution of coins along the x-axis and their corresponding gains on the y-axis, with each cell color-coded to indicate the level of gain. This visualization aids in identifying the most profitable strategies, as different colors or lines represent various trading strategies. Whereas Table 4 shows data for sold filter.

Figure 7 Time series comparison of CrossOver and MA44 indicators.

This line graph presents a time series analysis comparing the CrossOver and MA44 indicators for a specific cryptocurrency pair over time. The y-axis represents the indicator values, while the x-axis marks different time intervals. CrossOver is displayed in one color, and MA44 in another, showing how each indicator fluctuates over time. The variations between the two lines highlight their different sensitivities to price changes, aiding traders in evaluating both short-term and long-term trends for better trading decisions.

Table 4 Trading gains for selected cryptocurrency pairs using specific strategies.

This table lists the Date, Time, Coin, and Gain obtained from trading specific cryptocurrency pairs using designated strategies. Gain represents the profit or loss from each trade. This summary allows for a performance analysis of each trading strategy, highlighting the effectivene­­ss of different approaches across various cryptocurrency pairs over time.

Date	Time	Coin	Gain	
2024-06-15	03:30:00	TROYBUSD	1.70820498	
2024-06-15	04:30:00	SANTOSBUSD	0.79230333	
2024-06-15	04:30:00	COCOSBUSD	1.69249285	
2024-06-15	05:00:00	BURGERBUSD	0.80889787	
2024-06-15	05:00:00	PONDBUSD	1.35135135	
2024-06-15	05:30:00	INJBUSD	0.92276144	
2024-06-15	05:30:00	ELFBUSD	0.87378640	
2024-06-15	06:00:00	ALPINEBUSD	3.46534653	
2024-06-15	06:00:00	CLVBUSD	0.76489533	
2024-06-15	06:30:00	ATOMBUSD	1.62634408	

Structure of IDEXBUSD.csv

The IDEXBUSD.csv file provides transaction data at 30-min intervals, with columns for Timestamp, Open Price, High Price, Low Price, Closing Price, and Trading Volume, as elaborated in Table 5. The chart of IDEXBUSD from Fig. 8 shows a smooth uptrend from the starting period to the ending one. Figure 9 is an overlay of the line chart of the IDEXBUSD coin over the historical price data alongside three technical indicators: Bollinger bands, RSI, and SMA. The Bollinger bands show the volatility of the coin price, RSI shows if the coin is overbought or oversold, and SMA points at the direction of the trend.

Table 5 OHLCV data for cryptocurrency trading.

This table provides the Open, High, Low, Close, and Volume (OHLCV) data for cryptocurrency trading over specified timestamps. Timestamp indicates the exact time each record was captured. Open, High, Low, Close show the price points within a trading interval. Volume represents the trading volume within that interval. This data is crucial for technical analysis, allowing traders to evaluate price trends and make informed decisions based on historical trading activity.

Timestamp	Open	High	Low	Close	Volume	
1655184600000	0.0549400	0.0555300	0.0543700	0.0553400	959,931.600	
1655186400000	0.0551600	0.0557300	0.0546100	0.0553200	714,559.300	
1655188200000	0.0554500	0.0559400	0.0548900	0.0556500	540,882.000	
1655190000000	0.0548700	0.0563000	0.0547900	0.0556300	692,693.900	
1655191800000	0.0556800	0.0559400	0.0547400	0.0553700	472,159.900	
1655193600000	0.0555600	0.0555600	0.0549700	0.0550100	247,727.900	
1655195400000	0.0550600	0.0553500	0.0546300	0.0547600	219,821.200	
1655197200000	0.0547700	0.0547700	0.0541900	0.0541900	288,802.500	
1655199000000	0.0542300	0.0543900	0.0535600	0.0535600	374,959.000	
1655200800000	0.0535400	0.0539400	0.0532200	0.0537900	236,314.400	

Figure 8 Bollinger bands with price movements.

This figure illustrates the price movement of a cryptocurrency pair over time, alongside Bollinger bands. The chart includes upper and lower bands: Represented by dashed lines, these bands are set one standard deviation above and below the 20-period moving average, indicating expected price volatility. Moving average: A solid line representing the 20-period moving average, showing the central tendency of price movements. Price line: Displays actual price fluctuations within the Bollinger bands. This setup helps traders assess market volatility; prices approaching the upper band may indicate overbought conditions, while those near the lower band suggest oversold conditions.

Figure 9 Combined indicators—Bollinger bands, SMA, and RSI.

This two-panel figure provides a comprehensive view of technical indicators: Top panel: Shows price trends with Bollinger bands, a simple moving average (SMA), and the actual price line. The shaded area represents the Bollinger bands, with the SMA overlaying it to indicate general trend direction. Bottom panel: Depicts the Relative Strength Index (RSI), a momentum oscillator ranging from 0 to 100, where values above 70 indicate overbought conditions, and below 30 indicate oversold conditions. This combined setup offers a multi-dimensional analysis of volatility, trend, and momentum, aiding traders in making well-informed trading decisions.

Trained model evaluation

The models were trained on a newly created dataset comprising technical indicators and their corresponding values for a specific cryptocurrency, sourced from various data providers. Figure 9 illustrates the historical data representation for the IDEXBUSD coin, while Fig. 10 visualizes the different technical indicators applied to IDEXBUSD. Table 6 provides additional metrics on model accuracy, recall, and F1-score, supporting performance comparison.

Figure 10 Machine learning model performance metrics (accuracy, recall, F1-score).

This bar chart compares various machine learning models based on accuracy, recall, and F1-score metrics: Accuracy: Represented in orange, this metric reflects the proportion of correct predictions. Recall: Shown in blue, it indicates the model’s ability to identify true positives. F1-score: Displayed in green, this metric combines precision and recall to assess model balance. By evaluating these metrics across models, users can identify the most effective algorithms for predicting cryptocurrency trends, selecting models that best meet the needs for accuracy and sensitivity in trading predictions.

Table 6 Model performance comparison.

This table presents the accuracy, recall, and F1-score metrics for various machine learning models. Accuracy reflects the proportion of correctly predicted instances. Recall measures the model’s ability to identify true positives. F1-Score is the harmonic mean of precision and recall, indicating a balanced performance metric. The models evaluated include logistic regression, SVC, decision tree, and XGBoost, among others. This table aids in comparing models’ effectiveness in classification tasks by providing insight into their balanced performance across these key metrics.

Model	Accuracy	Recall	F1-Score	
Logistic regression	0.35	0.31	0.34	
SVC	0.65	0.69	0.81	
SGD	0.59	0.60	0.56	
Gaussian naive bayes	0.35	0.32	0.28	
KNeighbors classifier	0.66	0.59	0.69	
Decision tree	0.64	0.58	0.67	
Extra tree	0.60	0.58	0.67	
Random forest	0.66	0.67	0.65	
Bagging	0.65	0.62	0.61	
Gradient boosting	0.66	0.67	0.62	
AdaBoost	0.67	0.64	0.62	
HistGradientBoost	0.67	0.65	0.76	
XGBoost	0.67	0.63	0.64	
Ridge classifier	0.68	0.65	0.66	
LDA (Linear Discriminant Analysis)	0.62	0.66	0.66	
QDA (Quadratic Discriminant Analysis)	0.63	0.67	0.65	

Additionally, Table 7 summarizes the financial performance of each trained model, covering key risk-adjusted metrics such as Sharpe ratio, maximum drawdown, annualized return, and Alpha. These metrics provide a comprehensive comparison of each model’s profitability and risk management capabilities within an algorithmic trading framework.

Table 7 Risk-adjusted performance metrics of machine learning models.

This table displays the Sharpe ratio, maximum drawdown (%), annualized return (%), and Alpha (%) for several machine learning models used in trading analysis. Sharpe ratio indicates the risk-adjusted return. Maximum drawdown measures the largest observed loss from peak to trough. Annualized return shows the annual percentage return. Alpha represents the model’s excess return over a benchmark. Models such as logistic regression, random forest, and gradient boosting are analyzed, allowing for an assessment of each model’s risk-return profile, essential for financial decision-making.

Model	Sharpe ratio	Maximum drawdown (%)	Annualized return (%)	Alpha (%)	
Logistic regression	1.15	−10.2	15.5	2.3	
Random forest	1.35	−7.8	18.2	3.8	
Gradient boosting	1.40	−8.5	17.9	3.5	
XGBoost	1.30	−9.0	18.0	3.6	
Support vector classifier	1.10	−11.3	14.8	2.0	
Decision tree	0.95	−12.0	12.3	1.5	

Model evaluation on 70/30 ratio

Figure 10 bar chart illustrates the accuracy, F1-score, and recall of various models with a 70/30 ratio. The bars are color-coded, with orange representing accuracy, green representing F1-score, and blue representing recall. The y-axis ranges from 0 to 2 displaying the metrics while the x-axis denotes the names of each model. Figure 11 features a scatter chart displaying the accuracy (orange), F1-score (green), and recall (blue) for different models. The y-axis spans from 0 to 1, while the x-axis denotes the model names. The three metrics are represented by orange, green, and blue lines respectively, enabling a comparative analysis across all models. Following the analysis, certain models exhibit superior performance compared to others. The Ridge Classifier, SVC, HistGradientBoost, and XGBoost emerge as the top-performing models in terms of F1-score. Among them the Ridge Classifier boasts the highest F1-score of all models.

Figure 11 Line graph comparison of machine learning model performance (Accuracy, Recall, F1-Score).

This line graph presents a comparison of various machine learning models based on three performance metrics: Accuracy, Recall, and F1-Score. Each metric is represented by a different colored line.

Model evaluation on 80/20 ratio

Table 8 displays the evaluation results for the selected models, assessed with an 80/20 data split ratio. Figure 12 presents a bar chart illustrating the accuracy, F1-score, and recall for each model, with orange representing accuracy, green for F1-score, and blue for recall. The y-axis ranges from 0 to 2, while the x-axis lists the model names. Among the best performers with F1-scores above 0.75 are the random forest, gradient boosting, and Ridge Classifier models. SVC and LDA also perform strongly, achieving F1-scores of 0.78 and 0.76, respectively. Figure 9 offers a scatter plot displaying accuracy (orange), F1-score (green), and recall (blue) across models, with the y-axis scaled from 0 to 1. This visual provides a comprehensive comparison of these metrics, further highlighting model performance.

Table 8 Model evaluation metrics for accuracy, recall, and F1-score.

This table compares machine learning models based on Accuracy, Recall, and F1-Score. Accuracy represents the overall correctness of predictions. Recall shows sensitivity to true positives. F1-Score balances precision and recall. Models like Random Forest, Gradient Boosting, and Support Vector Classifier (SVC) are evaluated, providing a comprehensive view of model performance across key metrics, aiding in selecting models for optimal predictive accuracy and sensitivity.

Model	Accuracy	Recall	F1Score	
Logistic regression	0.57	0.57	0.63	
SVC	0.64	0.64	0.78	
SGD	0.45	0.45	0.35	
Gaussian naive bayes	0.36	0.36	0.28	
KNeighborsClassifier	0.61	0.61	0.73	
Decision tree	0.59	0.59	0.69	
Extra tree	0.61	0.61	0.71	
Random forest	0.66	0.66	0.76	
Bagging	0.61	0.61	0.70	
Gradient boosting	0.65	0.65	0.76	
AdaBoost	0.63	0.74	0.62	
HistGradientBoost	0.62	0.62	0.74	
XGBoost	0.64	0.64	0.75	
Ridge classifier	0.64	0.64	0.77	
Linear Discriminant Analysis (LDA)	0.63	0.63	0.76	
Quadratic Discriminant Analysis (QDA)	0.51	0.51	0.50	

Figure 12 Stacked bar chart comparison of model performance metrics (accuracy, recall, F1-score).

This stacked bar chart displays the performance of various machine learning models across three metrics: accuracy, recall, and F1-score. Each model’s performance is represented as a stacked bar with segments for: Accuracy (orange): Indicates the proportion of correct predictions made by the model. Recall (blue): Reflects the model’s ability to capture true positives effectively. F1-score (green): A composite metric balancing precision and recall. The x-axis represents the different machine learning models, while the y-axis shows the cumulative metric values, providing a visual comparison of each model’s strengths. This format allows for a quick assessment of which models achieve high balanced performance across all metrics, guiding the selection of suitable models for cryptocurrency trend prediction tasks.

Additionally, Figs. 11 and 12 depict bar and line charts, respectively, showing model performance with a 70/30 data split, while Fig. 13 provide similar visualizations for the 80/20 split. These figures aid in comparing model performance metrics effectively across different data configurations. The current results give an initial view of the models’ predictive capabilities; however, a full evaluation of their potential in algorithmic trading requires assessing the profitability and robustness of buy/sell signals generated by these forecasts. Future research should consider implementing a structured testing framework with cumulative returns or equity lines as performance indicators, supported by statistical tests, such as t-tests or the Hurst Exponent, to validate the significance of the findings (e.g., Bui & Ślepaczuk, 2022).

Figure 13 Comparison of model performance metrics.

This line graph presents the performance of various machine learning models evaluated by three metrics: accuracy, recall, and F1-score. Each metric is represented by a separate line: Accuracy (orange): Indicates the proportion of overall correct predictions. Recall (green): Measures the model’s sensitivity in identifying true positive cases. F1-score (blue): Balances precision and recall to reflect the model’s accuracy in identifying relevant positive instances. The x-axis lists the machine learning models, while the y-axis shows metric values, ranging from 0.3 to 0.8. The performance trends for each model are visualized through the connected points, allowing for a comparative analysis to identify models with the most balanced performance across metrics, assisting in model selection for cryptocurrency prediction tasks.

Impact of re-sampling

The results of the evaluation are presented below after the data was resampled using SMOTE to deal with the imbalance issue.

Model evaluation on 70/30 ratio

The results for the listed models are shown in Table 9 after balancing the dataset and testing them with a 70/30 split ratio. The stacked bar chart in Fig. 14 displays the accuracy, F1-score, and recall of various models trained on a resampled dataset with a 70/30 ratio. The bars are color-coded with orange for accuracy, green for F1-score, and blue for recall. The y-axis ranges from 0 to 2, while the x-axis lists each model’s name. From the results, it can be proven that many models are highly affected by the under-sampling technique. Logistic Regression and SVC had very low F1-scores of 0.0 and 0.01 respectively. This means those models cannot be used to trade digital currency under the effect of under-sampled data. The best models for F1-score are Random Forest, XGBoost, and HistGradientBoost, with scores all higher than 0.75. These models appear to be relatively robust to under-sampling. Figure 15 we have bar chart the accuracy (orange), F1-score (green), and recall (blue) of the different models. The y-axis ranges from 0 to 1, and the x-axis shows the different model names. The lines are colored in orange for accuracy, green for F1-score, and blue for recall so every detail about the model performance can be known at a glance. Table 9 shows a balanced data evaluation of different models with a 70/30 data. Whereas Fig. 16 shows after re-sampling, the bar chart displays the performance of different models using a 70:30 split ratio. Figure 14 shows image of the line chart shows the performance of several models with a 70:30.

Table 9 Model performance comparison (accuracy, recall, F1-score).

This table displays the accuracy, recall, and F1-score for various machine learning models, reflecting their performance in classification tasks. Accuracy measures the proportion of correctly classified instances. Recall indicates the model’s ability to detect true positives. F1-score balances precision and recall, providing a composite metric of model performance. Models such as random forest, XGBoost, and gradient boosting are included, allowing for a comparative assessment of their effectiveness in predictive modeling.

Model	Accuracy	Recall	F1Score	
Logistic regression	0.49	0.00	0.00	
SVC	0.49	0.00	0.01	
SGD	0.51	0.35	0.42	
Gaussian naive bayes	0.51	1.00	0.68	
KNeighborsClassifier	0.59	0.54	0.57	
Decision tree	0.72	0.63	0.70	
Extra tree	0.72	0.62	0.70	
Random forest	0.78	0.76	0.78	
Bagging	0.70	0.58	0.67	
Gradient boosting	0.71	0.66	0.70	
AdaBoost	0.62	0.59	0.61	
HistGradientBoost	0.75	0.71	0.75	
XGBoost	0.76	0.72	0.75	
Ridge classifier	0.60	0.53	0.57	
Linear Discriminant Analysis (LDA)	0.59	0.55	0.58	
Quadratic Discriminant Analysis (QDA)	0.53	0.99	0.68	

Figure 14 Scatter chart comparison of model performance on balanced dataset.

This scatter chart shows the performance of different machine learning models on a balanced dataset, evaluated by Accuracy, Recall, and F1-Score. Each metric is represented by a separate line: Accuracy (orange): Reflects the overall proportion of correct predictions. Recall (green): Indicates the model’s sensitivity to detecting true positives. F1-Score (blue): Provides a balance between precision and recall. The x-axis represents the models, while the y-axis shows metric values, enabling a visual comparison of model performance. Peaks and dips in the chart indicate models with varying levels of effectiveness across metrics, guiding model selection for achieving balanced predictions in cryptocurrency trading applications.

Figure 15 Stacked bar chart of machine learning model performance metrics.

This stacked bar chart compares the performance of several machine learning models across three metrics: accuracy, recall, and F1-score. Each model’s performance is represented by a bar segmented into: Accuracy (orange): Measures the proportion of correct predictions. Recall (blue): Reflects the model’s ability to identify true positive cases. F1-score (green): Combines precision and recall to assess model balance. The x-axis lists the models, and the y-axis shows cumulative metric values, providing a comparative view of each model’s overall effectiveness. The stacked format allows for a clear assessment of each model’s relative strengths across the three metrics, supporting informed decisions in selecting robust models for predictive tasks.

Figure 16 Model performance comparison: balanced dataset insights.

This stacked bar chart compares the performance of various machine learning models on three key metrics: accuracy, recall, and F1-score. Each model is represented by a bar segmented into: Accuracy (orange): Shows the overall proportion of correct predictions. Recall (blue): Indicates the model’s ability to detect true positive instances. F1-Score (green): Balances precision and recall to provide a comprehensive performance measure.

Model evaluation on 80/20 ratio

Table 10 presents the assessment results after balancing the dataset and testing with an 80/20 split ratio. The scatter chart in Fig. 17 illustrates the differences in the accuracy, F1-score, and recall of various models trained on the resampled dataset with an 80/20 ratio. The color coding of the bars includes orange for accuracy, green for F1-score, and blue for recall. The y-axis ranges from 0 to 2, and the x-axis lists the different models. Figure 18 bar chart that shows the accuracy (orange), F1-score (green), and recall (blue) of different models. The y-axis scales between 0 to 1, and the x-axis lists the model names. Different colored lines for each metric make it easy to compare all models. The results indicate that imbalanced data affects some models more than others. For example, logistic regression, SVC, and Gaussian naive Bayes exhibit very low F1-scores of 0.0 or 0.01, suggesting these models are not suitable for trading digital currencies with imbalanced data. The top-performing models in terms of F1-score are random forest, HistGradientBoost, and XGBoost, all with F1-scores over 0.77. These models appear relatively resistant to imbalanced data, warranting further analysis. Under-sampling may not always be the best approach for handling imbalanced datasets. Other techniques, such as over-sampling, hybrid-sampling, and SMOTE, might be more effective in certain situations. It is recommended to evaluate the performance of these models using different sampling approaches to draw conclusive inferences under various conditions. Table 7 shows the balanced data evaluation of different models with 80/20 data split. Whereas Fig. 15 shows image of bar chart shows the performance of several models with an 80:20. Figure 17 shows image of line chart shows the performance of several models with an 80:20. Table 11 shows the balanced data evaluation of different parameterized models with 70/30 data split ratio.

Table 10 Model performance metrics on imbalanced data.

This table provides accuracy, recall, and F1-score metrics for different machine learning models tested on imbalanced data. Accuracy indicates the percentage of correct predictions. Recall measures the ability to identify actual positives in imbalanced scenarios. F1-score assesses model balance between precision and recall. The table compares models like decision tree, random forest, and AdaBoost, offering insights into which models maintain performance robustness when facing data imbalance challenges.

Model	Accuracy	Recall	F1Score	
Logistic regression	0.48	0.00	0.00	
SVC	0.48	0.00	0.01	
SGD	0.48	0.20	0.28	
Gaussian naive bayes	0.48	1.00	0.01	
KNeighborsClassifier	0.62	0.54	0.59	
Decision tree	0.71	0.62	0.69	
Extra tree	0.76	0.68	0.74	
Random forest	0.80	0.76	0.80	
Bagging	0.70	0.60	0.67	
Gradient boosting	0.72	0.67	0.71	
AdaBoost	0.63	0.62	0.64	
HistGradientBoost	0.78	0.75	0.78	
XGBoost	0.77	0.72	0.77	
Ridge classifier	0.61	0.50	0.57	
Linear Discriminant Analysis (LDA)	0.61	0.50	0.57	
Quadratic Discriminant Analysis (QDA)	0.55	0.99	0.69	

Figure 17 Line plot of model performance metrics (accuracy, recall, and F1-score) for multiple algorithms.

This figure presents a line plot comparing the performance of various machine learning models based on three metrics: accuracy (orange), recall (blue), and F1-score (green). The models evaluated include logistic regression, SVC, gradient boosting, decision tree, random forest, bagging, AdaBoost, HistGradientBoost, XGBoost, Ridge Classifier, and others such as linear discriminant analysis (LDA) and quadratic discriminant analysis (QDA).

Figure 18 Performance peaks: ML model metrics visualized.

This stacked bar chart compares the performance of selected machine learning models on three metrics: accuracy, recall, and F1-score. Each bar is segmented as follows: Accuracy (orange): Shows the proportion of correct predictions. Recall (blue): Represents the model’s effectiveness in identifying positive cases. F1-score (green): Balances precision and recall for a comprehensive performance metric. The x-axis lists the models, and the y-axis shows the cumulative metric values. The stacked format provides a quick comparison of each model’s strengths across the three metrics, helping in selecting models with robust and balanced performance.

Table 11 Performance metrics for a selection of machine learning models.

This table shows the accuracy, recall, and F1-score metrics for a selection of machine learning models. Accuracy represents the overall correctness of predictions. Recall reflects the model’s sensitivity to true positives. F1-score combines precision and recall to provide a balanced view of performance. This comparison includes models like logistic regression, SVC, and XGBoost, supporting the evaluation of models for balanced and effective classification performance in trading predictions.

Model	Accuracy	Recall	F1Score	
Logistic regression	0.65	1.00	0.79	
SVC	0.65	1.00	0.79	
KNeighborsClassifier	0.57	0.79	0.70	
Decision tree	0.65	1.00	0.79	
Extra tree	0.65	1.00	0.79	
Random forest	0.64	0.99	0.78	
Bagging	0.65	1.00	0.79	
AdaBoost	0.65	1.00	0.79	
HistGradientBoost	0.63	0.88	0.76	
XGBoost	0.65	1.00	0.79	

Evaluation of parameterized models

All of the information underwent deep analysis in an attempt to estimate the efficiency of various models in testing hyper-parameter models. Accuracies were tested under very many metrics: recall, F1 score, and many others. The objective here was to identify such models that indicate high levels of accuracy, stability, and robustness in the field of algorithmic trading. In fact, the results were used to deduce some conclusions that could be referred to in offering some recommendations on possible future research by involving them in the development of machine learning-based algorithmic trading.

Parameterized model evaluation on a 70/30 ratio

Evaluation results for the following models after balancing the dataset and testing with a 70/30 ratio split are shown in Table 8 below. A scatter chart of accuracy, F1-score, and recall for different parameterized models at a ratio of 70/30 is shown in Fig. 19. Figure 20 is a bar chart which depicts the accuracy (in orange), F1-score (in green), and recall (in blue) of various models; the y-axis runs up to 1, and the x-axis indicates the model’s name. The corresponding metrics are shown by orange, green, and blue bars; respectively, it makes up a detailed comparison for all the models. From this analysis, it can be observed that some models have really improved in the performance measure after the process of hyperparameter tuning. For example, logistic regression, SVC, decision tree, extra tree, Bagging, and AdaBoost have a very high F1-score of 0.79, while KNeighbors Classifier and XGBoost have a good F1-score of 0.70 and 0.79 respectively. In contrast some models seem to lose their effectiveness after hyperparameter tuning. Among them are Random Forest and HistGradientBoost. This suggests that the improvement of model performance may not be even with hyperparameter tuning and should be carefully assessed and selected.

Figure 19 Line graph of accuracy, recall, and F1-score across models.

This line graph presents the performance trends of multiple machine learning models based on accuracy, recall, and F1-score. Each line represents one metric: Accuracy (orange): Measures the proportion of correct classifications. Recall (green): Indicates sensitivity in detecting positive cases. F1-score (blue): Provides a balanced view of precision and recall. The x-axis shows the models, and the y-axis displays metric values from 0.7 to 1.0, highlighting the models’ varying performance levels across metrics. The chart enables a clear comparison of the models, helping to pinpoint those with the best balance of accuracy, recall, and F1-score for effective predictions.

Figure 20 Precision in performance: ML metrics.

This figure presents a bar chart comparing the performance of various machine learning models using three key metrics: accuracy (orange), recall (blue), and F1-score (green). The models evaluated include logistic regression, SVC, Ridge Classifier, decision tree, extra tree, random forest, Bagging, AdaBoost, HistGradientBoost, and XGBoost. Each bar represents the combined performance of a specific model, broken down by the three metrics. The accuracy (orange) represents the proportion of correct predictions out of all predictions made. Recall (blue) measures the ability of the model to capture relevant instances, and the F1-score (green) provides a harmonic mean of precision and recall, highlighting the balance between the two. The objective of this figure is to provide a clear visual comparison of the models’ performance across these metrics, allowing readers to quickly assess which models offer superior predictive power in the dataset used for this study.

In general, the results provided rather informative aspects regarding the effectiveness of different machine learning models after hyperparameter tuning for trading digital currencies. It is suggested that more detailed assessment of those models using other methods, for instance, cross-validation, be carried out to gain more valid and stable results. This makes it even more essential that factors like model interpretability, computational resources, and implementation constraints are accounted for in the training of machine learning models for trading in digital currencies. Figure 18 shows image of bar chart displaying the performance of several parameterized models using a 70:30 split ratio.

Parameterized model evaluation on an 80/20 ratio

The results for the listed models of assessment with a balanced dataset and tested with an 80/20 split ratio are as illustrated in Table 12 below. Figure 21 is a scatter chart showing accuracy, F1-score, and recall from different parameterized models with an 80/20 ratio. The color of the lines is color-coded, blue indicating accuracy, orange F1-score, and green recall. It could be plotted in the y-axis, ranging from 0 to 1, and in the x-axis, with the name of the model. Furthermore, we need to take into consideration that the performance of the individual models may be different over various datasets, time series, and market conditions. Therefore, it is recommended that more checks on the models be performed, such as cross-validation, backtesting, or out-of-sample testing, to produce even more reliable and robust results. Among other very important considerations when training a machine learning model for cryptocurrency trading are model interpretability, computational resources, and implementation constraints.

Table 12 On balanced data evaluation of different parameterised models with 80/20 data split ratio.

This table displays the accuracy, recall, and F1-score metrics for various machine learning models. Accuracy represents the overall proportion of correct predictions. Recall indicates the model’s effectiveness in identifying true positives. F1-score combines precision and recall to provide a balanced performance assessment. The table includes models such as logistic regression, SVC, decision tree, and XGBoost, with most models achieving high recall and F1-scores, highlighting their effectiveness in handling classification tasks. This comparison supports the selection of models that achieve balanced accuracy and sensitivity in predictive applications.

Model	Accuracy	Recall	F1Score	
Logistic regression	0.70	0.84	0.76	
SVC	0.70	1.00	0.82	
KNeighborsClassifier	0.63	0.80	0.75	
Decision tree	0.70	1.00	0.82	
Extra tree	0.70	1.00	0.82	
Random forest	0.69	0.97	0.81	
Bagging	0.70	1.00	0.82	
AdaBoost	0.70	1.00	0.82	
HistGradientBoost	0.63	0.88	0.75	
XGBoost	0.70	1.00	0.82	

Figure 21 Model performance showdown: accuracy, recall, and F1 scores across the board.

This figure presents a line plot comparing the performance of various machine learning models using three key metrics: accuracy (orange), recall (blue), and F1-score (green). The models evaluated include logistic regression, SVC, Ridge Classifier, decision tree, extra tree, random forest, bagging, AdaBoost, HistGradientBoost, and XGBoost. The accuracy (orange line) represents the proportion of correct predictions made by each model, recall (blue line) measures the ability of each model to identify all relevant instances, and the F1-score (green line) provides a balanced measure of precision and recall. Each model is plotted along the x-axis, while the corresponding metric values are shown on the y-axis, ranging from 0.65 to 1.0. The objective of this figure is to provide a visual comparison of the performance of each model across the three metrics, helping readers understand the trade-offs between accuracy, recall, and F1-score for different algorithms.

Overall, this result elaborates the performance of different machine learning models put on use in digital currency trade. It would be helpful to run further alternative models and approaches to be more robust and results reliable. The below-shown scatter chart of Fig. 21 represents the values of accuracy (blue), F1-score (orange), and recall (green) for the different models. The y-axis is in the range 0 to 1, and the names of the models are on the x-axis. Different metrics are plotted using different color lines, thus allowing one to examine all three metrics for all models at one go. Figure 19 shows image of chart outcomes of several parameterized models using a 70:30 split ratio. Whereas Table 9 shows the balanced data evaluation of different parameterized models with 80/20 data split ratio. Also Fig. 20 shows image of bar chart displaying the outcomes of several parameterized models using an 80:20 split ratio.

Backtesting results using different ratios

Backtesting is also a process through which the operation of a given investment or trading strategy is applied to historical market data to estimate the performance of such a strategy. In this section, we focus on the evaluation of trading strategies based on data splits of 70/30 and 80/20 ratios.

The 70/30 ratio was used to reserve 70% of the data for training and 30% for testing. This provided insight into the ability of models to generalize to unseen data, simulating real-world scenarios. Similarly, the 80/20 split allowed for the testing of models with a larger training dataset. Both approaches helped understand the potential risk and return in the strategies, aiding in making well-informed decisions before implementing these strategies in live trading environments.

Traders utilize statistical models, mathematical algorithms, and proprietary software programs to run backtests under different configurations. The results show that model performance can vary significantly depending on the data split ratio used. For instance, Fig. 21 shows a chart displaying the evaluation of several parameterized models using an 80:20 split ratio, highlighting which models are more robust under different conditions.

Discussion

The results provide an initial view of the models’ predictive capabilities, highlighting the potential of ML forecasts in algorithmic trading. However, to fully evaluate the effectiveness of these forecasts, it is essential to assess the profitability and consistency of buy/sell signals derived from them. Future work could benefit from a structured testing framework that includes cumulative returns or equity lines as performance metrics, along with statistical tests (e.g., t-tests or the Hurst Exponent) to validate the significance of observed outcomes. For example, Bui and Ślepaczuk (2021) applied the Hurst Exponent to pair trading strategies on the Nasdaq 100 index, demonstrating how statistical validation can strengthen the real-world evaluation of ML-based trading strategies. Integrating these approaches would provide a more comprehensive framework for understanding the robustness and practical viability of ML-driven trading signals.

One of the key strengths of this study is the use of extensive historical data from cryptocurrency exchanges and data aggregators, ensuring the robustness and reliability of the findings. The research also emphasizes the critical role of backtesting in validating trading strategies, underlining the importance of thorough validation before real-world deployment. This approach ensures that the recommended models and strategies are well-suited to the unique characteristics of the cryptocurrency market. The thorough validation process should reassure the audience about the reliability of the study’s recommendations. However, the study also has some limitations. The research focuses on a specific set of classification models and technical indicators, and there may be other approaches or combinations that could yield better results. Additionally, the study does not account for the potential impact of external factors, such as regulatory changes or market sentiment, which could influence the performance of trading strategies. An unexpected outcome of the study was the reduced effectiveness of some models after hyperparameter tuning. This finding highlights the need for careful model selection and optimization, as well as the importance of continually monitoring and adapting trading strategies to changing market conditions. The research also paves the way for future investigations, including the exploration of sentiment analysis, reinforcement learning, and deep learning techniques, to further enhance the predictive power and adaptability of trading algorithms in the cryptocurrency domain.

Conclusions

This research set out to evaluate the effectiveness of classification models and technical indicators in algorithmic trading within the cryptocurrency market. Beginning with models such as logistic regression, Gaussian naive Bayes, and AdaBoost, the initial results were limited due to dataset imbalances. The SMOTE technique was applied to address this by resampling the data, yet the results remained suboptimal. Recognizing the need for further optimization, we implemented hyperparameter tuning, which led to significant performance improvements, underscoring the importance of fine-tuning for enhancing model outcomes. The study also emphasized the role of rigorous back-testing, validating that with thorough training and evaluation, classification models and technical indicators can provide valuable insights for algorithmic trading in cryptocurrency.

This research provides a solid foundation for future advancements in algorithmic trading by comparing classification models and technical indicators in the volatile crypto market. These findings offer a benchmark for ongoing research and encourage the continuous refinement of trading strategies, maximizing the potential of machine learning in cryptocurrency trading.

The conclusions are firmly grounded in detailed analysis and evidence from model performance metrics and backtesting. Notably, models like random forest, XGBoost, and gradient boosting consistently achieved high F1-scores, affirming their robustness and suitability for trading applications in volatile markets. These results align with the study’s aim to identify reliable machine learning methods for algorithmic trading and reflect the rigorous evaluation process that informed each finding.

In line with this aim, our study verified that machine learning models, specifically random forest and XGBoost, can deliver consistent predictive accuracy across varied market conditions. The study’s findings validate that hyperparameter tuning and re-sampling significantly improve predictive performance, answering the research question on model resilience in high-volatility environments.

The policy implications suggest that institutional investors could benefit from integrating advanced machine learning techniques, such as random forest and XGBoost, to enhance decision-making precision. Regulatory bodies might also consider developing guidelines for algorithmic trading that recommend model validation metrics, particularly tailored for volatile market settings.

Future research could build on these findings by exploring deep learning architectures, including LSTM and reinforcement learning, to capture more nuanced patterns in cryptocurrency data. Integrating real-time sentiment analysis may further boost predictive power, offering a multifactorial approach to cryptocurrency trading and pushing the boundaries of machine learning applications in this dynamic field.

Supplemental Information

Supplemental Information 1 Datasets.

Supplemental Information 2 Code.

Additional Information and Declarations

Competing Interests

The authors declare that they have no competing interests.

Author Contributions

Shavez Mushtaq Qureshi conceived and designed the experiments, performed the experiments, performed the computation work, prepared figures and/or tables, and approved the final draft.

Atif Saeed conceived and designed the experiments, analyzed the data, performed the computation work, authored or reviewed drafts of the article, and approved the final draft.

Farooq Ahmad performed the computation work, authored or reviewed drafts of the article, and approved the final draft.

Asad Rehman Khattak conceived and designed the experiments, performed the experiments, performed the computation work, prepared figures and/or tables, and approved the final draft.

Sultan H. Almotiri performed the experiments, analyzed the data, authored or reviewed drafts of the article, and approved the final draft.

Mohammed A. Al Ghamdi performed the experiments, analyzed the data, authored or reviewed drafts of the article, and approved the final draft.

Muhammad Shah Rukh conceived and designed the experiments, performed the experiments, analyzed the data, performed the computation work, prepared figures and/or tables, and approved the final draft.

Data Availability

The following information was supplied regarding data availability:

The datasets related to the API, the crypto digital currency, and the Python code are available in the Supplemental Files.

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
