# Peer review of "Evaluating machine learning models for predictive accuracy in cryptocurrency price forecasting"

_PeerJ Computer Science, doi:10.7717/peerj-cs.2626_

## Round 0.1 · original submission · Major Revisions

Dear authors,

Thank you for submitting your article. Based on reviews' comments, your article has not yet been recommended for publication in its current form. However, we encourage you to address the concerns and criticisms of the reviewers and to resubmit your article once you have updated it accordingly. Before submitting the paper, following should also be addressed:

1. Pay special attention to the usage of abbreviations. Spell out the full term at its first mention, indicate its abbreviation in parenthesis and use the abbreviation from then on. For example see the wrong usage for "ML" and "machine learning" in the manuscript
2. All of the values for the parameters of all algorithms selected for comparison should be provided.
3. The real contributions should be listed in the manuscript.
4. The current Introduction is simple and misses many contents related to the problem formulation. There is not a clear categorization of related work. Sections are confusing. "Materials and Methods" section is presented before "Related Work" section.
5. Introduction and Related Work sections seem broad, voluminous, and heterogeneous. You are supposed to focus on the main topic of the study and present a Literature Review in the form of tables in order to make research gaps and innovations easy to detect. Authoritative synthesis assessing the current state-of-the-art is absent.
6. Reviewers have advised you to provide specific references. You are welcome to add them if you think they are useful and relevant. However, you are under no obligation to include them, and if you do not, it will not affect my decision.

Best wishes,

·

Basic reporting

Dear Authors,

My full review is attached in a separate PDF file.

Please read this review carefully and ensure all my comments are addressed in the new corrected version of this paper before its subsequent submission.

Kind regards,
Reviewer

Experimental design

Attachment

Validity of the findings

Attachment

Additional comments

Attachment

Reviewer 2 ·

Basic reporting

Abstract:
The abstract is overall well-structured but lacks some key results and implications. For instance, "Results highlight the need to account for class imbalance" can be enriched with a few words on how class imbalance affected all the models and what worked best. This work concludes the abstract with future avenues of research; it needs to have a solid concluding sentence that sums up the study's practical application or broader implications.
Introduction:
The introduction describes the volatile nature of cryptocurrency markets well; however, more emphasis needs to be placed on the novelty of this research. What gaps in the literature are being precisely approached? This may help enhance the study's justification. Some references to existing works are too general. Try to include more recent or specific studies related to the methodologies applied in this research, particularly applications of machine learning in finance.
Literature Review:
While this section is comprehensive, attempting a more systematic organization would be beneficial. For example, studies could be grouped under headings such as "Cryptocurrency Market Studies," "ML Model Studies," and "Algorithmic Trading Strategies." Even though several studies are mentioned here, it will help explain how the findings of the current study compare to or add to those of the already existing ones. It would also make clear how your research differs in terms of novelty.
In both the introduction and the literature review, you could strengthen the discussion about the volatility and machine learning applications in cryptocurrency trading by referencing studies that specifically address the limitations of existing trading models, propose novel solutions in this domain, and provide a more thorough background on the state-of-the-art in algorithmic trading strategies, particularly those focusing on the robustness and scalability of machine learning models in dynamic trading environments.
• Yin, L., Wang, L., Cai, Z., Lu, S., Wang, R., AlSanad, A.,... Zheng, W. (2024). DPAL-BERT: A Faster and Lighter Question Answering Model. Computer Modeling in Engineering & Sciences, 141(1), 771-786. doi: https://doi.org/10.32604/cmes.2024.052622
• Wang, P., Song, W., Qi, H., Zhou, C., Li, F., Wang, Y.,... Zhang, Q. (2024). Server-Initiated Federated Unlearning to Eliminate Impacts of Low-Quality Data. IEEE Transactions on Services Computing, 17(3), 1196-1211. doi: 10.1109/TSC.2024.3355188
• Yin, L., Wang, L., Lu, S., Wang, R., Ren, H., AlSanad, A.,... Zheng, W. (2024). AFBNet: A Lightweight Adaptive Feature Fusion Module for Super-Resolution Algorithms. Computer Modeling in Engineering & Sciences , 140(3), 2315-2347. doi: https://doi.org/10.32604/cmes.2024.050853
• Wang, B., Zheng, W., Wang, R., Lu, S., Yin, L., Wang, L.,... Chen, X. (2024). Stacked Noise Reduction Auto Encoder–OCEAN: A Novel Personalized Recommendation Model Enhanced. Systems, 12(6), 188. doi: https://doi.org/10.3390/systems12060188
• Zhang, X., Pan, W., Scattolini, R., Yu, S., & Xu, X. (2022). Robust tube-based model predictive control with Koopman operators. Automatica, 137, 110114. doi: https://doi.org/10.1016/j.automatica.2021.110114
• Fan, W., Wu, X., & He, Q. (2024). Digitalization drives green transformation of supply chains: a two-stage evolutionary game analysis. Annals of Operations Research. doi: https://doi.org/10.1007/s10479-024-06050-0
• Zhou, Z., Zhou, X., Qi, H., Li, N., & Mi, C. (2024). Near miss prediction in commercial aviation through a combined model of grey neural network. Expert Systems with Applications, 255, 124690. doi: https://doi.org/10.1016/j.eswa.2024.124690
• Tu, Y., Liu, R., & Li, H. (2022). The Development of Digital Economy and the Future of the Trade Union Law of the People's Republic of China. Journal of Chinese Human Resources Management, 13(2), 76-85. doi: 10.47297/wspchrmWSP2040-800507.20221302
• Luo, J., Zhuo, W., & Xu, B. (2023). A Deep Neural Network-Based Assistive Decision Method for Financial Risk Prediction in Carbon Trading Market. Journal of Circuits, Systems and Computers, 33(8), 2450153. doi: https://doi.org/10.1142/S0218126624501536

Materials and Methods:
While the machine learning models are well described, along with the technical indicators, some points need further explanation. For example, it would be good to specify, in an obvious way, the parameters utilized for the different models, such as XGBoost and Random Forest, to allow for their complete reproducibility; likewise, any software packages or tools that may be used in this study should also be declared, such as the versioning of scikit-learn utilized, to replicate results better. For example, selecting data aggregation platforms needs more explanation, such as why Binance and CoinMarketCap were chosen. Do they seem to provide higher-quality data compared with other sources? This will help in reinforcing the methodology.
Results and Discussion:
While various results of different models are presented, discussion can be elaborative by comparing the models in detail. You may want to discuss why some models outperformed others, for instance, and relate this to characteristics of cryptocurrency markets, such as high volatility or unpredictability. The figures and tables are informative, but further captions and discussion about what these visualizations imply would give more context to the reader over what these results mean. Also, you could elaborate a bit more on the practical implications of these findings. How would such models be employed for live trading, and what kind of tools are taken into consideration? This section could also use the opportunity to highlight backtesting's role in risk management.
Conclusion:
The conclusion needs more emphasis on real-world implications. How will your results impact future cryptocurrency trading or adopting a machine learning model used in finance? Based on your findings, consider providing more robust actionable recommendations to researchers and practitioners. For instance, what model would you suggest for which scenario with varying levels of market volatility?
General Comments:
The language is generally clear, but there are places where more polished phrasing would improve readability. The manuscript needs additional proofreading for minor grammatical errors.

Experimental design

No comment

Validity of the findings

No comment

Additional comments

No comment

---

## Round 0.2 · accepted · Accept

Dear Authors,

One reviewer has accepted the manuscript. The invitation to review the revised manuscript was not responded to by one of the previous reviewers. I have assessed the revision myself and, in my view, your paper is now sufficiently improved following the last revision. It is therefore ready for publication.

Best wishes,

Reviewer 2 ·

Basic reporting

Thank you for your thorough and diligent revisions based on the reviewers' comments. The updated manuscript reflects significant improvements in quality, clarity, and depth, addressing all the critical feedback provided during the review process.
The scientific contributions are now presented more cohesively, and the methodological rigour and clarity of results make this work a valuable addition to the field. The revisions have elevated the manuscript to a high standard of academic quality.
I am pleased to recommend this paper for publication. It meets the journal's requirements and provides meaningful insights into the subject matter. Congratulations on your excellent work.

Experimental design

No comment

Validity of the findings

No comment

Additional comments

No comment